# Comparative Study of Extracellular Proteolytic, Cellulolytic, and Hemicellulolytic Enzyme Activities and Biotransformation of Palm Kernel Cake Biomass by Lactic Acid Bacteria Isolated from Malaysian Foods

**DOI:** 10.3390/ijms20204979

**Published:** 2019-10-09

**Authors:** Fu Haw Lee, Suet Ying Wan, Hooi Ling Foo, Teck Chwen Loh, Rosfarizan Mohamad, Raha Abdul Rahim, Zulkifli Idrus

**Affiliations:** 1Institute of Tropical Agriculture, Universiti Putra Malaysia, Serdang 43400 UPM, Selangor, Malaysia; fuhawlee4132@yahoo.com (F.H.L.); zulidrus@upm.edu.my (Z.I.); 2Department of Bioprocess Technology, Faculty of Biotechnology and Biomolecular Sciences, Universiti Putra Malaysia, Serdang 43400 UPM, Selangor, Malaysia; wsy@live.co.uk (S.Y.W.); farizan@upm.edu.my (R.M.); 3Institute of Bioscience, Universiti Putra Malaysia, Serdang 43400 UPM, Selangor, Malaysia; raha@upm.edu.my; 4Department of Animal Sciences, Faculty of Agriculture, Serdang 43400 UPM, Selangor, Malaysia; 5Institute of Tropical Forestry and Forest Products, Universiti Putra Malaysia, Serdang 43400 UPM, Selangor, Malaysia; 6Department of Cell and Molecular Biology, Faculty of Biotechnology and Biomolecular Sciences, Universiti Putra Malaysia, Serdang 43400 UPM, Selangor, Malaysia; 7Halal Products Research Institute, Universiti Putra Malaysia, Serdang 43400 UPM, Selangor, Malaysia

**Keywords:** biomass, *Lactobacillus plantarum*, palm kernel cake, biotransformation, solid-state fermentation, extracellular hydrolytic enzymes

## Abstract

Biotransformation via solid state fermentation (SSF) mediated by microorganisms is a promising approach to produce useful products from agricultural biomass. Lactic acid bacteria (LAB) that are commonly found in fermented foods have been shown to exhibit extracellular proteolytic, β-glucosidase, β-mannosidase, and β-mannanase activities. Therefore, extracellular proteolytic, cellulolytic, and hemicellulolytic enzyme activities of seven *Lactobacillus plantarum* strains (a prominent species of LAB) isolated from Malaysian foods were compared in this study. The biotransformation of palm kernel cake (PKC) biomass mediated by selected *L. plantarum* strains was subsequently conducted. The results obtained in this study exhibited the studied *L. plantarum* strains produced versatile multi extracellular hydrolytic enzyme activities that were active from acidic to alkaline pH conditions. The highest total score of extracellular hydrolytic enzyme activities were recorded by *L. plantarum* RI11, *L. plantarum* RG11, and *L. plantarum* RG14. Therefore, they were selected for the subsequent biotransformation of PKC biomass via SSF. The hydrolytic enzyme activities of treated PKC extract were compared for each sampling interval. The scanning electron microscopy analyses revealed the formation of extracellular matrices around *L. plantarum* strains attached to the surface of PKC biomass during SSF, inferring that the investigated *L. plantarum* strains have the capability to grow on PKC biomass and perform synergistic secretions of various extracellular proteolytic, cellulolytic, and hemicellulolytic enzymes that were essential for the effective biodegradation of PKC. The substantial growth of selected *L. plamtraum* strains on PKC during SSF revealed the promising application of selected *L. plantarum* strains as a biotransformation agent for cellulosic biomass.

## 1. Introduction

The second-millennium biotechnology approaches of the biotransformation of agricultural biomass into various useful products mediated by hydrolytic enzymes [1] produced by various microorganisms are vital and desperately need to be developed to facilitate the environmental challenges that are currently being encountered worldwide. Increasing understanding of biotransformation and properties of hydrolytic enzymes has driven the development of appropriate biotechnological techniques that have tremendous impacts on environmental waste disposal [2], in addition to the economic growth by providing new employment opportunities. The current biotechnological strategies focus mainly on energy production (biofuel) [3,4], biopolymer production [5], food production, and other value-added products, such as enzymes, biological media, metabolites, and vitamins [6,7,8,9,10,11].

Agricultural biomasses, such as bagasse, straws and stems, cobs, fruit peels, and husks, are excellent carbon sources, attributing to their lignocellulosic content. However, a major cause of limited usage of agricultural biomasses is owing to their inherent recalcitrance of the orderly crystalline cellulose and highly branched hemicellulose composition. Cellulases and hemicellulases that secreted by microorganisms via the fermentation process are essential enzymes required for the biotransformation of agricultural biomasses to simple sugar. Cellulase has been extensively studied and reported for its vital role in the production of biofuel from lignocellulosic materials [12,13]. Cellulase (EC 3.2.1.4) is a group of enzymes comprising endo-glucanase, exo-glucanase, and β-glucosidase. Endo-glucanase hydrolyses β-1,4 glycosidic bonds of cellulose into oligosaccharide polymers, whereas exo-glucanase and β-glucosidase degrade oligosaccharide polymers and other short-chain polysaccharides into β-glucose as simple sugar. Hemicellulase is a mixture of enzymes that consists of mannanase, xylanase, mannosidases, arabinosidases, and polygalacturonate hydrolyase. Complete hemicellulose hydrolysis is far more complex than cellulose due to its heterogenous structure and the composition of its monomers. To break down lignocellulosic biomasses effectively, both cellulases and hemicellulases should present and act synergistically and concomitantly. The cellulases and hemicellulases produced by microorganisms have been reported to be secreted in free form or multi-enzyme complexes. Cellulosomes is a multi-enzyme complex that commonly produces by anaerobic cellulolytic bacteria [14]. It is a large enzyme-integration protein complex with multiple cohesion and dockerin modules that exhibits high efficiency in biomass degradation [15]. Various fungus species (*Trichoderma* sp. [16,17,18], *Aspergillus* sp. [19], *Phanercochaete chrysosporium* [20,21]) and bacterial species (*Bacillus* sp. [22], *Clostridium* sp. [23], *Pseudoxanthomonas* sp. [24], *Ochrobactrum* sp. [25,26], *Klebsiella* sp. [27], *Cellulomonas* sp. [28,29], *Acinetobacter* sp. [30], and *Pseudomonas* sp. [31]) have been reported for their cellulases and hemicellulases activities.

Biotransformation that mediated by microorganisms via fermentation is a promising approach to produce useful products from agricultural biomasses, such as single cell proteins [32], organic acids [33], ethanol [34], mushrooms [35], biobutanol [18], and enzymes [17,36,37]. Solid materials such as peels, husks, and cobs that act as solid support are generally employed in the solid-state fermentation (SSF) process that is commonly performed in the presence of very limited free water. SSF is favorable process that can be mediated by a great variety of microorganisms, such as fungus, yeast, and bacteria. Despite the effective biotransformation of lignocellulosic materials by microorganisms [38], SSF also provides alternative substrates simultaneously, which can be applied in food, drug, textile, and dye industries [39,40]. Moreover, SSF is also an effective way to produce alternative feed ingredients with improved nutrient quality for livestock industries [41,42,43,44,45]. However, the usage of fungus in the biotransformation process can have major drawbacks due to their mycotoxin productions and longer maturation period. Hence, there is an urge to explore alternative biotransformation agents that are safe, effective, and beneficial for the SSF process.

Generally, lactic acid bacteria (LAB) is a gram-positive, acid tolerant, and present as rod-shaped or spherical bacteria producing lactic acid as the major organic acid from carbohydrate fermentation. LAB are commonly found in fermented food, such as fermented sausage, fermented cassava, black or white glutinous fermented rice, tempeh (soybean cake) and fermented vegetable, all of which have been shown to exhibit several extracellular hydrolytic enzyme activates [46,47], such as β-glucosidase, β-mannosidase, and β-mannanase [48,49]. In addition, LAB are generally known as “generally recognized as safe” (GRAS) and “quantified presumption of safety” (QPS) due to their beneficial probiotic characteristics [50,51]. The postbiotic metabolites produced by *Lactobacillus plantarum* strains, one of the major and important species of LAB that employed in this study, have been extensively reported as a versatile health supplement for rat and livestock as a growth promoter to produce safer and healthier meat for consumers [52,53,54,55]. *Paenibacillus polymyxa* and other LAB, such as *Lactobacillus salivarius,* could reduce the crude fiber and improved nutrient digestibility of broiler fed with fermented lignocellulosic biomass [56,57,58,59].

Furthermore, *L. plantarum* strains that used in present study exhibited broad antimicrobial activity against various pathogenic microorganisms, such as *Listeria monocytogenes*, *Clostridium perfringens*, *Salmonella enterica,* and *Escherichia coli* [60], suggesting that they are safe and beneficial for the SSF process. Therefore, seven strains of *L. plantarum* isolated from Malaysian foods were determined for their extracellular hydrolytic enzyme activities (proteolytic, cellulolytic, and hemicellulolytic enzyme activities) under acidic to alkaline conditions in the current study. Subsequently, three selected *L. plantarum* strains were determined for their potential as a biotransformation agent for SSF of PKC cellulosic biomass. Scanning electron microscopy analyses were also performed to observe the survivability and attachment of selected *L. plantarum* strains, as well as the surface changes of treated PKC during the SSF of PKC. This is the first report for the concomitant production of a mixture of extracellular proteolytic, cellulolytic, and hemicellulolytic enzymes by *L. plantarum* strains that are active from acidic to alkaline pH conditions. The versatile feature of the extracellular hydrolytic enzyme activities of the studied *L. plantarum* strains would warrant broad applications as potential bio-transformation agents in respective industries.

## 2. Results

### 2.1. Extracellular Hydrolytic Enzyme Activities of Lactobacillus plantarum Strains

#### 2.1.1. Effect of pH on the Extracellular Hydrolytic Enzyme Activities of *Lactobacillus plantarum* Strains

The extracellular cellulases (endo-glucanase, exo-glucanase, and β-glucosidase), hemicellulases (xylanase and β-mannanase), and endoprotease were determined for seven *L. plantarum* strains in this study. Filter paper-ase (FPase), carboxymethylcellulase (CMCase), and β-glucosidase were conducted for endo-glucanase, exo-glucanase, and β-glucosidase, respectively, using cell free supernatant (CFS) produced by *L. plantarum* strains grown in de Man, Rogosa and Sharpe (MRS) medium. The results obtained in this study demonstrated that the specific extracellular cellulolytic, hemicellulolytic, and proteolytic activities of *L. plantarum* strains were varied considerably under three different pH (5.0, 6.5, and 8.0) conditions.

As for the extracellular cellulolytic enzyme activities, the highest FPase activity was observed at pH 5.0 generally, except for the TL1 strain, which only exhibited FPase activity at pH 6.5 and pH 8.0 (Figure 1a). In comparison, the highest FPase activity at pH 5.0 was noted for RG14 strain (0.27 U/mg), followed by RG11, I-UL4, RS5, RI11, and B4 strains. Interestingly, the RG11 strain only demonstrated FPase activity at pH 5.0 and no FPase activity was detected for the B4 strain at pH 6.5. As for CMCase activities under different pH conditions, as shown in Figure 1b, the RG14, TL1, and I-UL4 strains only demonstrated two CMCase activities, whereas the RG11 RI11, RS5, and B4 strains exhibited three CMCase activities. However, the RG11 strain exhibited the highest CMCase activity at pH 6.5 with 0.58 U/mg, whereas RG14, RI11, TL1, and B4 showed higher CMCase activity at pH 8.0 as compared to pH 5.0 and 6.5. Surprisingly, the tested *L. plantarum* strains exhibited much lower β-glucosidase enzyme activities as compared to FPase and CMCase activities and only β-glucosidase enzyme activity was detected for RG14, RG11, RI 11, and RS5 at pH 5.0, whereas the highest β-glucosidase enzyme activity (0.017 U/mg) was exhibited by the B4 strain at pH 8.0, as shown in Figure 1c. As for pH 5.0, the RS5 strain demonstrated the highest β-glucosidase activity with 0.014 U/mg. However, it was not significantly different from the β-glucosidase of RG14 (0.012 U/mg), RG11 (0.011 U/mg), and RI11 (0.012 U/mg) strains. Furthermore, the β-glucosidase activity of the RG14, RG11, and RI11 strains was not detected at pH 6.5 and pH 8.0 respectively.

As for the extracellular hemicellulolytic activities, three xylanase activities were detected at pH 5.0, 6.5, and 8.0 for the studied *L. plantarum* strains, except for the RG14 strain, which was only active at pH 5.0 and 6.5 (Figure 1d). As for pH 5.0, the RS5 strain exhibited the highest xylanase activity (0.17 U/mg) as compared to B4, I-UL4, RG11, RG14, RS5, TL1, and RI11 strains. Interestingly, the highest xylanase activity at pH 6.5 and pH 8.0 was noted for the RG14 (0.21 U/mg) and B4 (0.15 U/mg) strains, respectively. Generally, the β-mannanase activity of the tested *L. plantarum* strains was found to be most active at pH 5.0 in comparison to pH 6.5 and pH 8.0, and the I-UL4 strain showed the highest enzyme activities with 0.40 U/mg, followed by the TL1 strain with 0.32 U/mg, RG14 strain with 0.30 U/mg, RI11 strain with 0.23 U/mg, B4 strain with 0.13 U/mg, RS5 strain with 0.11 U/mg, and RG11 strain with 0.08 U/mg (Figure 1e). Although the RI11, B4, RG11, and RS5 strains exhibited β-mannanase activity at pH 6.5, the enzyme activity was not as active at pH 5.0.

All the studied *L. plantarum* strains showed endoprotease activity under broad pH conditions, as illustrated in Figure 1f, although the endoprotease activity was varied amongst the strains under different pH conditions. In comparison, the highest endoprotease activity was recorded by the B4 strain (3.14 U/mg) at pH 6.5, whereas the RI11 strain exhibited the highest endoprotease activity at pH 5.0 with 1.74 U/mg.

#### 2.1.2. Rating of the Overall Extracellular Hydrolytic Enzyme Activities of *Lactobacillus plantarum* Strains

The specific enzyme activities of extracellular cellulase (FPase, CMCase, β-glucosidase), hemicellulase (xylanase, β-mannanase), and endoprotease produced by all tested *L. plantarum* strains were rated and the total score of overall specific enzyme activities was calculated and presented in Figure 2. To determine the total score of overall specific enzyme activities, a score was given to each enzyme activity produced by each *L. plantarum* strain under acidic, near-neutral and alkaline pH conditions. The lowest enzyme activity was given score 1 and the highest enzyme activity was given score 7. Subsequently, the total score from all enzyme activities of each *L. plantarum* strain was calculated and compared by two-dimensional competitor analysis and presented as a radar chart of a web-like format in a full scale of 100, as shown in Figure 2. The axis coordinate is located at the center of the radar chart. The distance from the center point indicates the total score of the extracellular hydrolytic enzyme activities of the producer strain. The total score of extracellular hydrolytic enzyme activities for three different pH conditions has demonstrated that RG14 strain obtained the highest score (84), followed by the RI11 strain (76), RS5 strain (76), B4 strain (76), RG11 strain (73), I-UL4 strain (68), and TL1 strain (51). Therefore, the RG14, RG11, and RI11 strains were selected for subsequent experiment of PKC biotransformation via SSF.

### 2.2. Solid State Fermentation of Palm Kernel Cake

#### 2.2.1. Viable Cell Count of Fermented Palm Kernel Cake Extract

Figure 3 shows the LAB population (log CFU/mL) of PKC extracts treated with *L. plantarum* RG14, RG11, and RI11 for 14 day of incubation period. The fermented PKC samples were collected at two-day intervals. Similar to the initial LAB count (RG14, 6.26 log CFU/mL; RG11, 6.93 log CFU/mL and RI11, 7.06 log CFU/mL) was observed for the PKC extract obtained from the treated PKC by RG14, RG11, and RI11 strains via SSF at the end of the 14-day incubation period. However, no LAB was detected for the untreated PKC extract (negative control) throughout the 14 days of incubation. The viable LAB population was increased by 15.51%, 17.03%, and 27.37%, respectively, for RI11, RG11, and RG14 treated PKC extracts from Day 0 to Day 2 of incubation. The LAB population of treated PKC extracts of RI11 and RG11 strains was subsequently decreased to the initial viable cell count at Day 6 and remained constant from Day 6 to Day 14 of incubation. Nevertheless, the treated PKC extract of strain RG14 showed the highest LAB count from Day 2 to Day 6, before it decreased to approximately the initial viable cell count at Day 12 in comparison to strains RI11 and RG11. Therefore, all three selected LAB strains could adapt, survive, and grow well on PKC under SSF condition for 14 days of incubation under limited free water availability condition.

#### 2.2.2. Reducing Sugar Concentration of Fermented Palm Kernel Cake Extract

The reducing sugar concentration of the treated PKC extract of RG14, RG11, and RI11 strains was lower than the untreated PKC extract, as shown in Table 1. The sugar concentration (mg/mL) declined drastically from Day 0 to Day 2 (*p* ˂ 0.05), whereby the sugar concentration of treated PKC extract of strain RI11 declined by 32.29% at Day 2. However, the treated PKC extract of strains RG11 and RG14 showed lower reducing sugar concentration at Day 0 and further decreased by 67.80% and 62.30%, respectively, after Day 2 (*p* ˂ 0.05) of incubation. Interestingly, the reducing sugar concentration of PKC treated by strains RG11 and RG14 was maintained from Day 2 to Day 14 in comparison.

#### 2.2.3. Solubilised Protein Concentration of Fermented Palm Kernel Cake Extract

The solubilised protein concentration (mg/mL) of untreated and treated PKC extracts are shown in Table 2. Generally, the solubilised protein concentration of treated PKC extracts was lower than the untreated PKC extract from Day 4 to Day 14 of incubation. However, the solubilised protein concentration of untreated PKC extract (control) and treated by strains RG14, RG11, and RI11 were increased by 25.20%, 6.57%, 14.09%, and 15.77%, respectively, although the solubilised protein concentration were not significantly different (*p* ˂ 0.05) between the treated and untreated PKC at Day 14 of incubation.

#### 2.2.4. Hydrolytic Enzyme Activities of Fermented Palm Kernel Cake Extract

Figure 4 illustrates the cellulases [filter paper-ase (FPase), avicelase, carboxymethylcellulase (CMCase), β-glucosidase], hemicellulases (β-mannanase, xylanase), and proteolytic (endoprotease) enzyme activities of PKC extracts treated with *L. plantarum* strains (RG14, RG11, and RI11) via SSF.

As for the cellulases activities, FPase was not detected for both untreated and treated PKC extracts (results not shown). Nevertheless, the treated PKC extract of strains RG14, RG11, and RI11 showed avicelase activity throughout the 14 days of incubation (Figure 4a). No significant increase in avicelase activity was found in the untreated PKC extract throughout the incubation period, except at Day 12 of incubation. The treated PKC extract of isolate RG14 exhibited the highest avicelase activity (0.75 U/mg) at Day 12 of incubation, followed by the RI11 and RG11 strains. Interestingly, the occurrence of a few maximum avicelase activities of treated PKC extracts were noted, despite different activities being recorded throughout the incubation periods for the treated PKC extracts, whereby three maximum avicelase activities were noted for RG14 treated PKC extracts at Day 2, Day 8, and Day 12, whereas three maximum avicelase activities of RG11 treated PKC extracts were noted at Day 2, Day 6, and Day 12, respectively.

Despite the CMCase activity of all treated PKC extracts being detected throughout the 14 days of incubation period, the maximum CMCase activity was detected at Day 6 for the treated PKC extracts of both strains of RI11 (0.71 U/mg) and RG11 (0.67 U/mg) (Figure 4b). Nevertheless, the highest CMCase activity was demonstrated by the treated PKC extract of RG14 at Day 4 (0.84 U/mg). Interestingly, the maximum CMCase activity of all treated PKC extracts occurred between Day 4 and Day 6, which was between the first and last maximum avicelase activities of Day 2 and Day 12, respectively.

Generally, the β-glucosidase activity of the treated PKC extracts was not detected at Day 2 of incubation, thus indicating that the enzyme activity was not produced at the early stage of the SSF process, as shown in Figure 4c. The β-glucosidase activity of the treated PKC extracts was the lowest as compared to other extracellular hydrolytic enzyme activities, despite the maximum β-glucosidase activity of the RG11 treated PKC extracts (0.26 U/mg) and RG14 treated PKC extracts (0.18 U/mg) being exhibited at Day 4 and the maximum β-glucosidase activity of treated PKC extract of RI11 (0.12 U/mg) and control being noted at Day 6. In comparison to avicelase activity, most of the maximum β-glucosidase activity occurred mainly at the mid-point of SSF incubation period (Day 4 to Day 6), which was coincidence with the maximum CMCase activities of treated PKC extracts.

Two maximum activities of hemicellulolytic enzyme activities were detected for RI11 and RG11 treated PKC extracts throughout the 14 days of the SSF period (Figure 4d). Interestingly, the β-mannanase enzyme activity of the treated PKC extracts was the highest among the extracellular hydrolytic enzyme activities. The maximum β-mannanase activity of the treated PKC extract of RG11 was noted at Day 6 (1.68 U/mg) and it was the highest enzyme activity in comparison to other strains.

The xylanase activity of the treated PKC extracts of strains RG14, RG11, and RI11 also exhibited similar trends to β-mannanase activity, whereby more than one maximum xylanase activity was detected for the treated PKC extracts throughout the incubation period (Figure 4e). The maximum xylanase activity of the PKC extract treated with strains RI11, RG11, and RG14 was occurred at Day 2 (0.47 U/mg), Day 4 (0.48 U/mg), and Day 12 (0.51 U/mg), respectively.

Endoprotease is among the vital enzymes when it comes to endo-chain protein biodegradation. Figure 4f shows that a few maximum endoprotease activity of the treated PKC extract was detected between Day 2 and Day 12 of incubation, whereby the maximum endoprotease activity was demonstrated by the treated PKC extracts of strain RG14 (6.03 U/mg) at Day 8 of incubation. In contrast, the maximum endoprotease activity was occurred at Day 4 for the treated PKC extract of strain RG11 (4.59 U/mg) and Day 8 for the treated PKC extract of strain RI11 (4.70 U/mg).

#### 2.2.5. Attachment and Growth of *Lactobacillus plantarum* Strains on Palm Kernel Cake

The scanning electron microscopy observation of the untreated and treated PKC did not show the presence of any bacteria in the initial period of SSF (data not shown). Figure 5 shows that the selected *L. plantarum* strains began to attach to the PKC surface on Day 4 (as shown by arrow indications) of SSF. The selected *L. plantarum* strains were attached around the grooved, porous, and uneven surfaces of PKC with appendages of extracellular polysaccharides matrices formed around the cell surface (Figure 6 with arrows indication) on Day 6, that were observed under higher magnification of scanning electron microscopy analyses. The extracellular polysaccharide matrices allowed them to hold and attached well to the PKC surface.

## 3. Discussion

### 3.1. Extracellular Hydrolytic Enzyme Activities of Lactobacillus plantarum Strains

#### 3.1.1. Effect of pH on Specific Extracellular Hydrolytic Enzyme Activities of *Lactobacillus plantarum* Strains

LAB are known to produce lactic acid from carbohydrate fermentation, both in aerobic and anaerobic conditions. Therefore, LAB are a popular choice in food industries, due to their shelf life extension ability by creating acidic condition in fermented food with the presence of lactic acid. Despite fungi and cellulolytic-degrading *Bacillus* sp., the concurrent occurrence of multi extracellular hydrolytic enzyme productions, including cellulolytic, hemicellulolytic, and proteolytic enzymes, have not been reported elsewhere for LAB. Furthermore, it has been reported that fungi and *Bacillus* sp. are effective bio-degraders for cellulosic materials and hence they have been studied extensively for various applications [22,48,61,62,63,64,65,66,67]. In the current study, we explored the potential of the *L. plantarum* (one of the major and important species of LAB) strains as versatile multi extracellular hydrolytic enzyme producers. Indeed, the results of the present study have clearly shown the occurrence of a cocktail of multi extracellular cellulolytic, hemicellulolytic, and proteolytic enzyme activities of *L. plantarum* strains, which are essential and vital for the degradation of complex non-starch polysaccharide and protein component of cellulosic biomass.

The extracellular cellulase (FPase, CMCase, and β-glucosidase) activities were observed for all the studied *L. plantarum* strains under broad pH conditions, but with different specific enzyme activities. However, most of the *L. plantarum* strains showed higher extracellular enzymatic activity under acidic conditions (Figure 1), which was between pH 5.0 to 6.5, indicating that the extracellular hydrolytic enzymes produced by *L. plantarum* strains were active and have the potential to degrade fibrous and crystalline cellulosic materials into simple sugar under acidic condition, such as PKC, since the pH of PKC is near-acidic condition. The feature of the cocktail mixture of extracellular cellulosic enzyme activities was mostly reported for fungus [68,69] and other cellulose-degrading bacteria [70]. Despite the proteolytic activity of LAB being been reported extensively, the occurrence of a cocktail mixture of cellulosic enzyme activity profiles and their respective hydrolytic activities has not been reported elsewhere for LAB or *L. plantarum* strain. The biodegradation of cellulosic materials by anaerobic cellulose-degrading bacteria has been reported to be attributed to the presence of extracellular protein complexes known as cellulosome [14], that consist of primary, anchoring, and adaptor scaffoldin protein molecules. The primary scaffoldin contains a cellulose-binding module that targets the cellulosome complex to the plant cell wall [15]. The versatile cocktail mixture of cellulosic enzyme profiles of the studied *L. plantarum* strain resembles the feature of cellulosome, exhibiting different cellulolytic enzyme activities under broad pH conditions that are essential for the efficient degradation of cellulosic materials.

Similar reports have made by Mawadza et al. and Yasemin et al., whereby CMCase activity produced by *Cellulomonas* sp., *Bacillus* sp., and *Micrococcus* sp. was more active under acidic to neutral conditions [62,63]. Generally, acidophilic enzymes show higher activity under acidic conditions [63], due to the presence of proton-translocating ATPase that are responsible for pH homeostasis against low pH conditions [64,71]. Nevertheless, the hydrolysis of cellulobiose and cellodextrin into simple sugar by β-glucosidase of LAB was recently reported [72,73,74]. Different LAB isolates produced different β-glucosidase with different optimal pH conditions, as clearly demonstrated in current study.

Two types of hemicellulolytic enzyme activities, namely xylanase and β-mannanase, were determined in this study. Both enzymes play an important role in hydrolysing the complex xylan and mannan of non-starch polysaccharide into xylose and mannose, respectively. Most of the xylanolytic and mannolytic enzymes were found in hemicellulolytic degrading bacteria and fungi [75,76,77]. Interestingly, xylanase produced by the studied *L. plantarum* strains was active under broad pH conditions, which was from pH 5.0 to pH 8.0, whereas *L. plantarum* TL1 exhibited higher enzyme activity when pH increased. The results obtained were similar to the findings of Scheirlink et al., who reported that xylanase activity was stable above pH 5.0 and the enzyme was irreversible inactivated below pH 4.5 [70]. Nevertheless, xylanase activity produced by *Bacillus* sp. was more active at neutral to alkaline conditions [78,79,80,81]. Therefore, *L. plantarum* has the ability to produce xylanase for the biodegradation of xylanolytic materials (Figure 1) under broad pH condition.

Different types of β-mannanase have different physical characteristics and exhibit different maximal enzyme activity under different pH conditions. Most of the β-mannanase that produced by the studied *L. plantarum* strains were found to be most active at pH 5.0. Certain LAB have the ability to degrade mannan as prebiotic molecules that have a stimulating function for the growth of LAB [44,45]. The β-mannanase and β-mannosidase produced by *Enterococcus casseliflavus* FL2121 were more active between pH 6.0 to pH 7.4 [45]. The results of this study were partially in agreement with those of Nadaroglu et al. [48], who investigated the clarification of fruit juice by using purified mannanase enzyme from *L. plantarum* M24. Purified β-mannanase produced by *Bacillus stearothermophilus* showed dimeric enzyme activity, with the maximal activity occurring between pH 5.5 and 7.5 [74,82].

LAB are known to possess a proteolytic system [83,84] to degrade the complex protein molecules to develop flavor and to enhance the nutrient digestibility of fermented foods. Endoproteolytic enzyme activity produced by the selected *L. plantarum* strains in the present study was also detected under broader pH conditions, which is in agreement with the extracellular protease activities of LAB isolated from Malaysian foods [85]. Lim et al. [86] also reported the comparative studies of versatile extracellular proteolytic activities of LAB and their potential for extracellular amino acid productions as feed supplements. In addition, the starch digestibility of corn silage could be improved further by proteases from LAB and *Bacillus subtilis* [87,88].

It was clearly shown in this study that the tested *L. plantarum* strains have the capability to produce a versatile extracellular hydrolytic enzyme cocktail that active under broad pH conditions. Therefore, it is worthwhile to perform a purification procedure via liquid chromatography, such as Fast Protein liquid chromatography or LC-MS/MS analyses to characterise the cellulolytic enzymes cocktail in detail.

#### 3.1.2. Rating of Extracellular Hydrolytic Enzyme Activities of *Lactobacillus plantarum* Strains

Three *L. plantarum* strains were subsequently selected based on the overall score of the extracellular hydrolytic enzyme activities (Figure 2), namely, RI11, RG11, and RG14 strains. They were selected for the biotransformation of PKC via SSF since they exhibited the highest score for the overall extracellular hydrolytic enzyme activities. The presence of multi extracellular hydrolytic enzyme activities synergistically are essential and important for the efficient degradation of cellulosic, hemicellulosic, and proteinaceous materials. The versatile cocktail mixture of extracellular hydrolytic enzyme activities warrants a broad application of the selected *L. plantarum* strains as an effective biotransformation agent for cellulosic biomasses.

### 3.2. Solid State Fermentation of Palm Kernel Cake

#### 3.2.1. LAB Viable Count of Fermented Palm Kernel Cake Extract

The results of the SSF of PKC demonstrated that the LAB population of treated PKC increased during the early stage (Day 2–Day 6) of SSF, but the LAB population declined significantly (*p* ˂ 0.05) after Day 6 (Figure 3). The LAB population increased by approximately 1 log CFU/mL, despite the reducing sugar concentration being decreased tremendously during the early stage of the SSF, as shown in Table 1, indicating that the selected LAB strains could degrade and utilise the nutrients available in PKC [89] for their growth. In addition, the loss of moisture content in PKC during SSF might affect the sustainability of bacteria growth. It was reported that the water activity required for the optimal growth of *L. plantarum* was more than 0.93. The bacteria growth will be retarded if the water activity is lower than 0.91 [90,91]. The introduction of 100% (*v*/*w*) moisture level allowed the selected *L. plantarum* strains to survive throughout the 14 days of SSF of PKC in the current study. The production of extracellular hydrolytic enzymes by the selected *L. plantarum* strains during SSF of PKC was noted in concerted manners for simultaneous and efficient protein, hemicellulose, and cellulose biodegradation of PKC.

#### 3.2.2. Solubilised Protein Concentration of Fermented Palm Kernel Cake Extract

The solubilised protein concentration of the treated PKC extract was lower than the untreated PKC extract, as shown in Table 2. However, the treated PKC extract exhibited a slow increase in protein concentration until the end of the SSF period. Extracellular protein biosynthesis or solubilisation of PKC protein by the selected *L. plantarum* strains might explain the increasing soluble protein concentration throughout the 14 days of SSF period [92]. The extracellular cellulolytic (Avicelase, CMCase and β-glucosidase), hemicellulolytic (β-mannanase, xylanase), and proteolytic enzyme activities of strains RI11, RG11, and RG14 were substantiated and verified again using the treated and untreated PKC extracts.

#### 3.2.3. Hydrolytic Enzyme Activities of Fermented Palm Kernel Cake Extract

The avicelase activity of the treated PKC extract was detected throughout the 14 days of SSF periods, with the most active enzyme activity occurring in the early and late stages of the fermentation period. This is the first report that has shown the studied *L. plantarum* strains have the ability to hydrolyse the exo-chain of crystalline cellulosic polymer. Such enzyme activity is believed to enhance the biodegradation efficiency of the cellulosic material present in PKC. The sporadic occurrences of a few maximum avicelase activities of the treated PKC extracts were most likely to be due to the synergistical effect with the presence of other hydrolytic enzyme activities that were essential for the effective biodegradation of cellulose polymer of PKC. High CMCase activity was also evident in the treated PKC extract. The crystalline cellulose of PKC was most likely to be degraded by CMCase to the oligo chain of cellulose polymers. Therefore, strains RI11, RG11, and RG14 were able to utilise the degraded cellulosic material as an energy source for their growth, as demonstrated by the evidence noted for the increased viable cell counts between Day 2 to Day 6 of SSF in this study. In comparison, the most active β-glucosidase activity responsible for reducing sugar liberation from cellobiose was exhibited between Day 4 and Day 6 of SSF. This finding was in agreement with Alshelmani et al. [55,93], who reported that high CMCase, xylanase, and mannanase activity was exhibited by *P. polymyxa*, *Bacillus megaterium,* and *Bacillus amyloliquefaciens* during Day 4 of SSF of PKC.

The main hemicellulosic component of PKC is mannan, which constitutes 57.8% of the total PKC composition [94]; hence the hemicellulose degradation ability of the selected *L. plantarum* strains was of prime importance to determine in this study. Selected strains RI11, RG11, and RG14 demonstrated β-mannanase activity at Day 6 and Day 8 of SSF. Certain manno-oligosaccharides that derived from PKC after the hydrolysis reaction of β-mannanase could serve as prebiotics for the growth of specific LAB [95]. The β-mannanase activity exhibited by the selected RI11, RG11, and RG14 strains could have contributed to the growth and sustainability of LAB population during the SSF of PKC. In addition, the LAB treated PKC can be applied as a promising alternative feed ingredient for livestock due to the fact that the bio-transformed insoluble linear mannan composition of PKC, which contains a very low degree of galactose residues, could be beneficial to the digestive tract of monogastric animals since it would not cause high viscosity in the gut content that may lead to the gut inflammation problem reported frequently in monogastric animals [96,97]. Similarly, the xylanase activity of the treated PKC extract was detected throughout the SSF of PKC. This is the first evidence of xylanase activity produced extracellularly by LAB. The production of xylooligossaccharides (XOS) from the degradation of complex xylan that attributed to the extracellular xylanase produced by selected LAB during the bio-transformation of PKC was beneficial [98,99,100], since they have been reported to be a prebiotic agent that could support the growth of LAB, specifically in the gut of monogastric animals [101]. According to Ahmed et al. [56], a significant reduction in the crude fiber of canola meal fermented by *L. salivarius* could improve the growth performance of broilers. Furthermore, adequate inclusion level of PKC fermented by *P. polymyxa* or *Paenibacillus curdlanolyticus* into broiler feed could improve body weight gain without adverse effects on the meat quality and growth performance of broiler as compared to unfermented PKC [57,58,59].

The protein content of PKC is between 14.50% and 19.24%, as reported by Ezieshi and Olomu [102], Ramachandran et al. [103], Alshelmani et al. [45], and Abdollahi et al. [104]. Endoprotease is essential to hydrolyse the complex protein of PKC into simple peptides to promote LAB growth [105]. The most active endoprotease activity was detected between Day 4 and Day 8 of SSF of PKC. It has been proposed that the entrapment of protein resides could occur in cellulosic material, and this would restrict the accessibility of hydrolytic enzymes [106]. The concomitant and synergistic productions of protease together with cellulolytic and hemicellulolytic enzymes extracellularly by selected *L. plantarum* strains enable the efficient biotransformation of PKC occurred in concerted manner to release peptides and sugars residues from cellulosic and hemicellulosic components of PKC. Therefore, the biotransformed PKC can be used as an alternative feed ingredient to support the growth of livestock. The sugar residues released from treated PKC could be a useful feedstock for bio-fuel production.

#### 3.2.4. Attachment and Growth of *Lactobacillus plantarum* Strains on Palm Kernel Cake

The growth capability of the selected strains RI11, RG11, and RG14 on PKC was further substantiated and verified by observing the treated and untreated PKC under scanning electron microscope. The purpose of this study was to reveal the PKC biodegradation, growth, and attachment of selected strains RI11, RG11, and RG14 on the PKC surface. No bacteria were observed on the PKC surface on Day 0 of the SSF. The PKC surface resembled a rough and uneven surface [107] under scanning electron microscope observation. Robust growth of selected strains RI11, RG11, and RG14 in the early stage of SSF was observed at Day 2, whereby a substantial number of bacterial cells were attached to the PKC surface. Most of the selected strains RI11, RG11, and RG14 were widely distributed over the grooved and uneven porous surfaces of PKC after Day 4 of SSF, indicating that they were actively grown on PKC (Figure 5 with arrow indications), following the biodegradation of fibrous materials of PKC, which supported well the growth of *L. plantarum* strains upon the synergistic actions of the excreted cocktail of extracellular hydrolytic enzyme mixtures, as shown in Figure 4.

Interestingly, the higher magnification micrographs taken for the Day 6 of fermented PKC samples revealed the formation of unique exopolysaccharide appendages surrounding the outer surface of RI11, RG11, and RG14 strains, which clearly allowed them to hold on and attached well to the PKC surfaces, as shown in Figure 6. The close contact of selected *L. plantarum* strains to the PKC surfaces would have facilitated the biodegradation process of PKC by the excreted cocktail of extracellular hydrolytic enzymes that would have reacted in the concerted manner during the SSF of PKC. In addition, more complex exopolysaccharides were observed on the surface of *L. plantarum* RI11, whereby the isolates underwent biofilm formation with the neighbor for cell attachment (Figure 6 with arrow indications). It has been shown that single culture or mixed cultures of LAB are able to form biofilm [108,109,110,111,112]. Transmission electron microscopy analysis should be conducted to characterise the membrane ultrastructure of the studied *L. plantarum* strains during the SSF of PKC.

Similarly, the formation of biofilm was reported for *Tepimicrobium xylanilyticum* BT14, *Lactobacillus* sp., and *Cellulomonas flavigena* [112,113,114]. The function of biofilm is to collect the nutrients from its neighbor through the bacteria culture in order to prevent the leaking of hydrolysis products to the environment and to reduce the competition for nutrients between other microorganisms [113]. Similar biofilm formation was most probably observed for the studied *L. plantarum* to prevent the leaking of hydrolysis products (sugars and amino acids) from PKC to sustain its growth throughout the SSF of PKC.

## 4. Materials and Methods

### 4.1. Microorganisms and Maintenance

Seven LAB (*L. plantarum* RI11, *L. plantarum* RG11, *L. plantarum* RG14, *L. plantarum* TL1, *L. plantarum* I-UL4, *L. plantarum* RS5, and *L. plantarum* B4) isolated from Malaysian foods were obtained from the Laboratory of Industrial Biotechnology, Department of Bioprocess Technology, Faculty of Biotechnology and Biomolecular Sciences, Universiti Putra Malaysia [115,116,117,118]. The bacterial cultures were maintained and revived as described by Foo et al. [118] and Moghadam et al. [119]. The bacterial cultures were maintained at −20 °C in MRS medium (Merck, Darmstadt, Germany) supplemented with 20% (*v*/*v*) glycerol.

### 4.2. Preparation of Extracellular Hydrolytic Enzymes

The active *L. plantarum* strain was washed once with sterile 0.85% (*w*/*v*) NaCl (Merck, Germany) solution and adjusted to 10^9^ CFU/mL to be used as inoculum. The working cultures of *L. plantarum* strains were prepared by inoculating 10% (*v*/*w*) of 10^9^ CFU/mL active bacterial cell into the MRS media and incubated at 30 °C for 10 h for each experiment, followed by centrifugation (Benchtop Microfuge 20R, Beckman Coulter, Indianapolis, USA) at 10,000 × g, 4 °C for 15 min. The CFS was then collected by filtration through cellulose acetate membrane (Sartorius Minisart, 0.22 µm, Göttingen, Germany) as described by Alshelmani et al. [93] and used for the determination of extracellular hydrolytic enzyme activities. The CFS was kept at −20 °C until enzyme assays were conducted.

### 4.3. Effect of pH on Extracellular Hydrolytic Enzyme Activities

Extracellular hydrolytic enzyme activities, including cellulase, hemicellulose, and endoprotease, were determined from acidic to alkaline pH conditions using three different buffer systems (pH 5.0, 0.1 M sodium acetate solution; pH 6.5, 0.1 M sodium phosphate buffer solution; pH 8.0, 0.1 M tris-hydrochloric acid buffer solution) for CFS prepared with MRS medium, whereas the extracellular hydrolytic enzyme activities of PKC extracts were determined at pH 5 only.

### 4.4. Cellulase and Hemicellulase Activities

Carboxymethylcellulase (CMCase), avicelase, filter paper-ase (FPase), xylanase, and β-mannanase activities were determined by using 2% (*w*/*v*) carboxymethylcellulose (Sigma, St. Louis, MO, USA), 0.5% (*w*/*v*) avicel pH10 (Fluka, Tokyo, Japan), 50 mg (1 cm × 6 cm) Whatman No.1 filter paper, 1% (*w*/*v*) xylan from birchwood (Sigma, St. Louis, MO, USA), 0.5% (*w*/*v*) locust bean gum from ceratonia silique seed (Sigma, St. Louis, MO, USA) as substrates [22,26,46,120,121,122]. Carboxymethylcellulose, avicel, filter paper, xylan, and locust bean gum were prepared using respective buffer solutions.

The enzyme assay mixtures, including enzyme blank (EB, respective mixture solution without enzyme), substrate blank (SB, respective mixture solution without substrate), and reaction mixture (RM, containing enzyme and substrate) were prepared and incubated at 37 °C for 60 min. The reduced sugar concentrations produced in the cellulase and hemicellulase assays were determined by the modified DNS (3,5-dinitrosalicyclic acid) method of Miller [123]. A volume of 100 µL DNS and 10 µL sodium hydroxide (0.1 M NaOH) was added to 100 µL assay mixture after 60 min of enzyme assay. The mixture was then boiled for 5 min. A volume of 800 µL deionised water (dH_2_O) was added to the assay mixture and left at room temperature for 20 min, prior to the determination of absorbance (Abs) at 540 nm (Cary 50 Probe UV-visible spectrophotometer, Agilent Technologies, Santa Clara, CA, USA). The mixture of DNS and NaOH solution was used as the reference solution for Abs determination (RA). Hence, the net Abs obtained from the assay mixture of EB, SB, and RM was subtracted from the RA. The calculation of net Abs was as below:

Net Abs = (RM−RA) − (SB-RA) − (EB−RA),

RM: Abs of reaction mixture; SB: Abs of substrate blank; EB: Abs of enzyme blank; RA: Abs of reference solution containing DNS and NaOH

One unit of specific enzyme activity (U/mg) was defined as the amount of reducing sugar produced in 1 min of reaction time by 1 mg of enzyme solution under assay condition. Glucose was used as a standard reference for CMCase and FPase determination, whereas mannose and xylose were used as a standard reference for the determination of xylanase and β-mannanase.

β-glucosidase activity was determined by using 5 nM 4-nitrophenyl-β-D-glucopyranoside as substrate prepared in the respective buffer solutions [43]. Assay mixtures of EB, SB, and RM were prepared and incubated at 37 °C for 30 min, followed by the addition of 500 µL 0.5 M sodium bicarbonate to the assay mixture. The Abs was then determined at 373 nm (Cary 50 Probe UV-visible spectrophotometer, Agilent Technologies, Santa Clara, CA, USA). Sodium bicarbonate solution was used as the reference solution for Abs determination (RA). Abs obtained from the assay mixture of EB, SB, and RM was then subtracted from the Abs of reference solutions. The calculation of net Abs is shown below:

Net Abs = (RM−RA) − (SB-RA) − (EB−RA),

RM: Abs of reaction mixture; SB: Abs of substrate blank; EB: Abs of enzyme blank; RA: Abs of reference solution containing sodium bicarbonate

One unit of β-glucosidase activity (U/mg) was defined as the nmole of *p*-nitrophenol released from 4-nitrophenyl-β-D-glucopyranoside per min of assay time by 1 mg of enzyme solution under assay condition. *p*-nitrophenol was used as the standard reference.

### 4.5. Endoproteolytic Activity

Endoprotease activity was determined by using 2.5% (*w*/*v*) sulphanilamide azocasein (Sigma Aldrich, St. Louis, MO, USA) as the substrate, which was mixed homogenously in the respective buffer solutions [117]. Assay mixtures were prepared and incubated at 37 °C for 30 min before 0.75 mL of 10% (*w*/*v*) trichloroacetic acid was added. The mixture was subsequently incubated at room temperature for 30 min to precipitate the substrate–enzyme complexes. The precipitates were removed by centrifugation at 12,000× g, 4 °C for 10 min. Then, 0.6 mL of clear supernatant was mixed with 0.6 mL of 0.1 M NaOH and incubated at room temperature for 15 min prior to Abs measurement at 450 nm. One unit of endoprotease activity (U/mg) was defined as the enzyme that capable of hydrolysing azocasein to produce 0.001 in Abs change per min of reaction time per mg of enzyme protein under assay condition.

### 4.6. Protein Concentration Determination

Protein concentration was determined by using the modified Lowry method [124] to calculate the specific enzyme activity of cellulase, hemicellulase and endoprotease. Bovine serum albumin (Sigma Aldrich, St. Louis, MO, USA) was used as standard reference.

### 4.7. Rating of Extracellular Hydrolytic Enzyme Activities of Lactobacillus plantarum Strains

All enzyme activities, including cellulase (FPase, CMCase, β-glucosidase), hemicellulase (xylanase, β-mannanase), and proteolytic (endoprotease) activities, were given a score in order to determine the highest hydrolytic enzyme producer among the seven *L. plantarum* strains. The score given was ranged from 1 (lowest) to 7 (highest), and the competitor analysis radar chart was plotted to compare the total score of hydrolytic enzyme activities among the studied *L. plantarum* strains by displaying the total score in a two-dimensional web-like graph. The axis coordinate was located at the center of the two-dimensional web-like plot. The distance from the coordinate indicated the total score of the enzyme activity, whereby the higher the score, the longer the distance from the coordinate center. The top three scores of *L. plantarum* strains were selected for subsequent biotransformation of PKC via SSF.

### 4.8. Solid State Fermentation of Palm Kernel Cake

PKC was ground to a particle size of 0.5 mm and sterilised at 121 °C for 15 min, followed by drying overnight in an oven to remove excessive moisture. Sterilised PKC (15 g) was added to 15 mL dH_2_O prior to inoculating with 10% (*v*/*w*) of 10^9^ CFU/mL active culture of *L. plantarum* RI11, *L. plantarum* RG11, and *L. plantarum* RG14. PKC without bacterial culture was used as the negative control. The PKC was incubated at 30 °C for 14 days, whereby sample was collected in two-day intervals.

At each sampling interval, 30 mL of dH_2_O was added into the respective fermented PKC to prepare the PKC extract. A volume of 1 mL PKC extract was used to determine the viable cell count (Log CFU/mL). The remaining PKC extract was further incubated in a rotatory shaker for 30 min at 30 °C and 130 rpm (Innovar 42 Incubator Shaker Series, New Brunswick Scientific, Hamburg, Germany). A clear PKC extract was obtained by centrifugation at 10,000× g, 4 °C for 15 min (Himac CR 22GII, Hitachi High Speed Refrigerated Centrifuge, Hitachi Koki Co Ltd., Tokyo, Japan). The PKC extract was later filtered through a cellulose acetate membrane (Sartorius Minisart, 0.22 µm, Göttingen, Germany) and kept at −20 °C for further analyses of reducing sugar concentration, solubilised protein concentration, cellulase, hemicellulose, and proteolytic enzyme activities. PKC pellets were kept for scanning electron microscopy analyses.

### 4.9. Lactic Acid Bacteria Viable Count of Fermented Palm Kernel Cake Extract

A volume of 1 mL PKC extract was 10-fold serial diluted using sterile 0.85% (*w*/*v*) NaCl (Merck, Darmstadt, Germany) solution. The diluted mixture was spread on MRS agar plates and incubated at 30 °C for 48 h. The viable cell count was expressed as log CFU/mL.

### 4.10. Scanning Electron Microscopy Analyses of Fermented Palm Kernel Cake

The surface structure of PKC was observed under scanning electron microscope (JOEL JSM 6400, Japan Inc., Tokyo, Japan) at the Microscopy Laboratory, Institute of Biosciences, Universiti Putra Malaysia, Selangor, Malaysia. PKC pellet was fixed in 2.5% (*w*/*v*) glutaraldehyde for 4–6 h at 4 °C, followed by washing with 0.1 M sodium cacodylate buffer (3 × 10 min) for primary fixation procedure [38]. The pre-fixed sample was then post-fixed in 1% (*w*/*v*) osmium tetraoxide (OsO_4_) for 2 h at 4 °C. The sample was then washed with 0.1 M sodium cacodylate buffer (3 × 15 min), followed by a dehydration procedure by using ascending grades of acetones: 10 min for 35%, 50%, 75% and 95%; and lastly 100% acetone solution for 3 × 15 min. The dehydrated sample was then subjected to critical point drying for 0.5 h prior to mounting on aluminium stub. The mounted sample was sputter coated (Baltech SCD 005 Sputter Coater, Liechtenstein, Scotia, New York, USA) with a thin layer of gold and observed under an operating voltage of 15 kV. Micrographs were digitally taken by using SemAfore 5.21 (software, Ahmedabad, Gujarat, India).

### 4.11. Statistical Analysis

The mean difference of the results obtained in each experiment was analysed using analysis of variance (ANOVA), followed by the Tukey post hoc test at the significant level of 0.05 (SAS 9.4 software, Cary, NC, USA).

## 5. Conclusions

The *L. plantarum* stains tested in the present study showed the production of a cocktail mixture of extracellular proteases, cellulases, hemicellulases, and endoprotease, all of which were active from acidic to alkaline conditions, inferring the versatile extracellular hydrolytic enzyme capacity of the studied *L. plantarum* strains. *L. plantarum* RI11, RG11, and RG14 were subsequently selected as the best candidates for the biotransformation of PKC via SSF based on their total score of extracellular hydrolytic enzyme activates. The substantial growth on PKC during the SSF, as shown by scanning electron microscopy analyses, warrants the broad application of the selected *L. plantarum* RI11, RG11, and RG14 as a promising biotransformation agent for cellulosic materials to be used as animal feeds and biofuel. Synergistic interactions between extracellular hydrolytic enzyme activities that reacted in the concerted manner were observed for the simultaneous and effective biodegradation of PKC components to sustain the growth and biosynthesis activities of the selected *L. plantarum* strains of RI11, RG11, and RG14. It is worth further characterising the extracellular hydrolytic cocktails by liquid chromatography, such as Fast Protein Liquid Chromatography and LC-MS/MS, in order to understand or to explore further the potential of extracellular hydrolytic enzyme cocktail produced by *L. plantarum* strains. Transmission electron microscopy analysis should be conducted to characterise the formation of unique exopolysaccharide appendages surrounding the outer surface of RI11, RG11, and RG14 strains during the SSF of PKC.

## Figures and Tables

**Figure 1 ijms-20-04979-f001:**
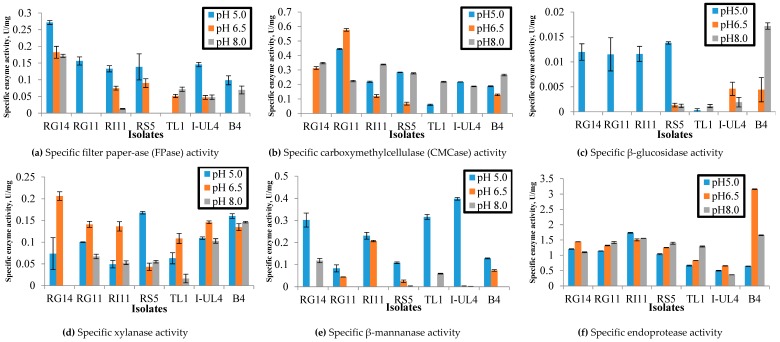
Effect of pH on specific extracellular hydrolytic enzyme activities of *Lactobacillus plantarum* strains. (**a**) Specific filter paper-ase activity (FPase); (**b**) Specific carboxymethylcellulase activity (CMCase); (**c**) Specific β-glucosidase activity; (**d**) Specific xylanase activity; (**e**) Specific β-mannanase activity; (**f**) Specific endoprotease activity. Notes: RG14, *L. plantarum* RG14; RG11, *L. plantarum* RG11; R111, *L. plantarum* RI11; RS5, *L. plantarum* RS5; TL1, *L. plantarum* TL1; I-UL4, *L. plantarum* I-UL4; B4, *L. plantarum* B4. Error bar: standard error of mean (SEM), *n* = 3.

**Figure 2 ijms-20-04979-f002:**
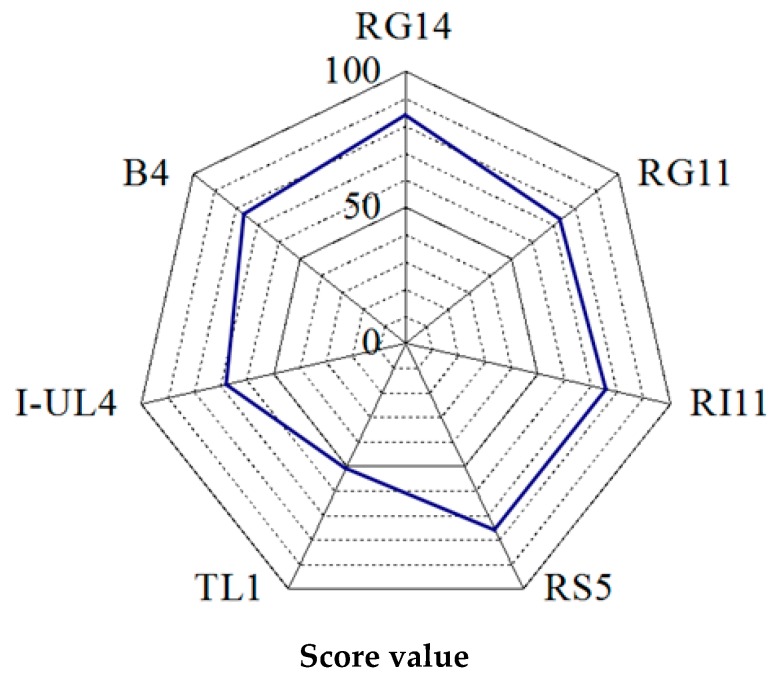
Rating of extracellular hydrolytic enzyme activities of *Lactobacillus plantarum* strains. Notes: RG14, *L. plantarum* RG14; RG11, *L. plantarum* RG11; R111, *L. plantarum* RI11; RS5, *L. plantarum* RS5; TL1, *L. plantarum* TL1; I-UL4, *L. plantarum* I-UL4; B4, *L. plantarum* B4.

**Figure 3 ijms-20-04979-f003:**
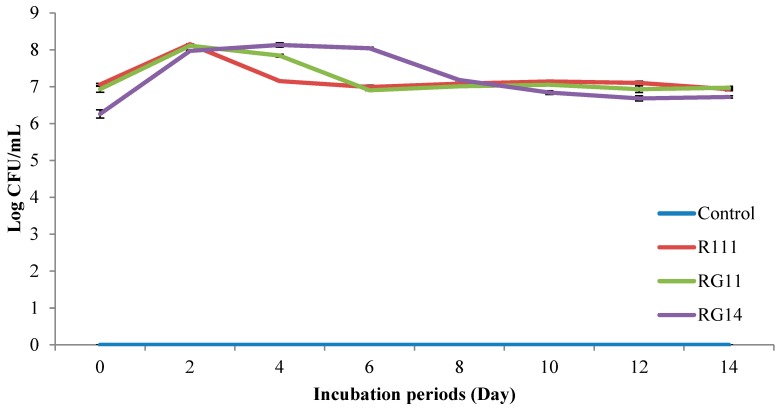
Lactic acid bacteria population of palm kernel cake extracts treated with *Lactobacillus plantarum* strains via solid state fermentation. Notes: RG14, *L. plantarum* RG14; RG11, *L. plantarum* RG11; R111, *L. plantarum* RI11; Control: untreated PKC extract. Error bar: standard error of mean (SEM), *n* = 3.

**Figure 4 ijms-20-04979-f004:**
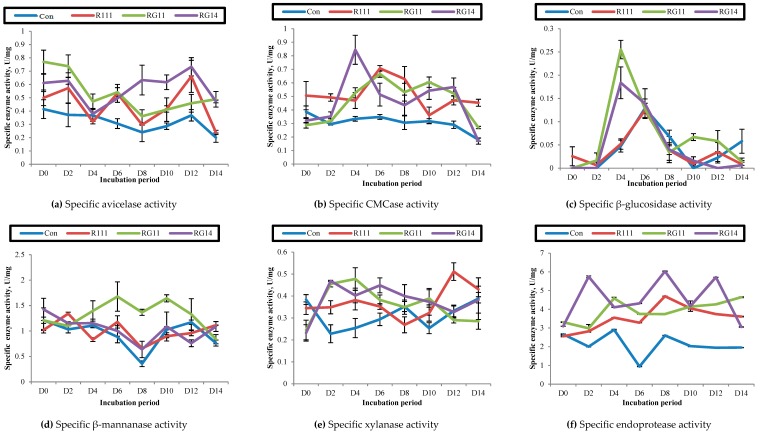
Cellulolytic, hemicellulolytic, and proteolytic enzyme activities of palm kernel cake extract treated with *Lactobacillus plantarum* R111, RG11, and RG14 via solid state fermentation for 14 days. (**a**) Specific avicelase activity; (**b**) Specific carboxymethylcellulase activity (CMCase); (**c**) Specific β-glucosidase activity; (**d**) Specific β-mannanase activity; (**e**) Specific xylanase activity; (**f**) Specific endoprotease activity. Notes: Con, unfermented palm kernel cake; R111, *L. plantarum* RI11; RG11, *L. plantarum* RG11 and RG14, *L. plantarum* RG14. Error bar: standard error of mean (SEM), *n* = 3.

**Figure 5 ijms-20-04979-f005:**
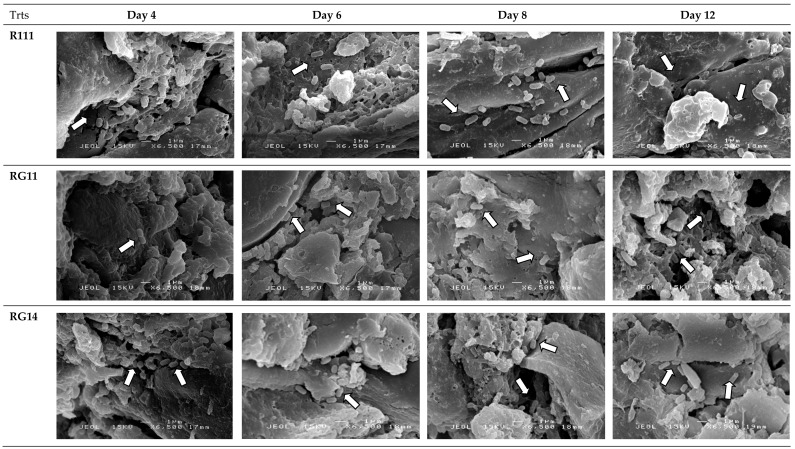
Scanning electron micrographs of treated palm kernel cake by selected *Lactobacillus plantarum* strains. The magnification was 6,500 x (Bar: 1µm). Abbreviations: R111, *L. plantarum* R111; RG11, *L. plantarum* RG11; RG14, *L. plantarum* RG14. White arrows show the attachment of *L. plantarum* strains around the grooved, porous, and uneven surfaces of PKC during the SSF of PKC biomass.

**Figure 6 ijms-20-04979-f006:**
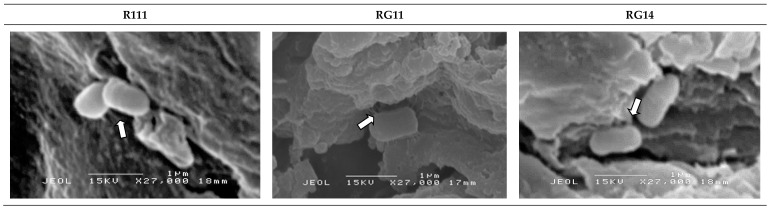
Scanning electron micrograph of exopolysaccharide formation of selected *Lactobacillus plantarum* strains on palm kernel cake at Day 6 of solid-state fermentation (Bar: 1 µm). Abbreviations: R111, *L. plantarum* R111; RG11, *L. plantarum* RG11; RG14, *L. plantarum* RG14. White arrows show the appendages of extracellular polysaccharides matrices formed around the cell surface of selected *L. plantarum* strains which allowed them to hold and attached well to the PKC surface during the SSF of PKC biomass.

**Table 1 ijms-20-04979-t001:** Reducing sugar concentration of untreated and treated palm kernel cake extracts.

Days	Reducing Sugar Concentration (mg/mL)			
	Control	R111	RG11	RG14
Day 0	0.49^cA^	0.68^aA^	0.59^aA^	0.61^aA^
Day 2	0.61^bcA^	0.44^bB^	0.19^dC^	0.23^bcC^
Day 4	0.62^bcA^	0.24^cdB^	0.21^cdB^	0.21^cB^
Day 6	0.67^abcA^	0.21^cdB^	0.29^bcB^	0.24^bcB^
Day 8	0.70^abcA^	0.16^dC^	0.28^bcB^	0.21^cBC^
Day 10	0.73^abcA^	0.24^cdBC^	0.25^bcdB^	0.20^cC^
Day 12	0.79^abA^	0.24^cdB^	0.30^bB^	0.23^bcB^
Day 14	0.93^aA^	0.27^cB^	0.30^bB^	0.31^bB^
SEM ±	0.05	0.06	0.04	0.05

Notes: Values bearing different superscripts (a–d) in a column indicate a significant difference (*p* < 0.05), while identical or common superscripts (a–d, ac) denote a non-significant difference (*p* > 0.05). Values bearing different superscripts (A–C) in a row indicate a significant difference (*p* < 0.05), while identical or common superscripts (A–C, AB) denote a non-significant difference (*p* > 0.05). Overall SEM ± was given by first calculating the sample mean and then calculating overall mean in a column or an individual parameter in different groups. Abbreviation: Control, unfermented palm kernel cake; R111, *L. plantarum* R111; RG11, *L. plantarum* RG11; RG14, *L. plantarum* RG14. *n* = 3.

**Table 2 ijms-20-04979-t002:** Solubilized protein concentration of untreated and fermented palm kernel cake extracts.

Days	Solubilized Protein Concentration (mg/mL)			
	Control	R111	RG11	RG14
Day 0	0.95^bA^	0.99^aA^	0.95^aA^	0.94^aA^
Day 2	0.98^bA^	1.00^aA^	0.86^aA^	0.92^aA^
Day 4	1.03^abA^	0.93^aA^	0.93^aA^	0.91^aA^
Day 6	1.09^abA^	1.00^aA^	1.05^aA^	0.99^aA^
Day 8	1.18^abA^	1.00^aA^	1.05^aA^	1.03^aA^
Day 10	1.20^abA^	1.00^aA^	1.05^aA^	1.00^aA^
Day 12	1.17^abA^	1.02^aA^	1.05^aA^	0.98^aA^
Day 14	1.27^aA^	1.07^aA^	1.10^aA^	1.11^aA^
SEM ±	0.04	0.01	0.03	0.02

Notes: Values bearing different superscripts (a–d) in a column indicate a significant difference (*p* < 0.05), while identical or common superscripts (a–d, ac) denote a non-significant difference (*p* > 0.05). Values bearing different superscripts (A–C) in a row indicate a significant difference (*p* < 0.05), while identical or common superscripts (A–C, AB) denote a non-significant difference (*p* > 0.05). Overall SEM ± was given by first calculating the sample mean and then calculating overall mean in a column for an individual parameter in different groups. Abbreviation: Control, unfermented palm kernel cake; R111, *L. plantarum* R111; RG11, *L. plantarum* RG11; RG14, *L. plantarum* RG14. *n* = 3.

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
