# Peer review of "Comparative Study of Extracellular Proteolytic, Cellulolytic, and Hemicellulolytic Enzyme Activities and Biotransformation of Palm Kernel Cake Biomass by Lactic Acid Bacteria Isolated from Malaysian Foods"

_ijms, 2019, doi:10.3390/ijms20204979_

Round 1
Reviewer 1 Report
Presented manuscript (ijms-58958) is an updated and improved version of the ijms-527050 manuscript and deals with production of the extracellular proteolytic, cellulolytic and hemicellulolytic enzymes form seven Lactobacillus plantarum strains isolated from Malaysian foods and production. In the study, obtained enzymes were extracted, characterized and compared for biotransformation of palm kernel cake biomass. Furthermore, also activity and amount of the produced enzymes was evaluated as well as effect of pH on this parameter was examined.
The topic discussed in the manuscript is definitely worth of investigation and is within the scope of the International Journal of Molecular Sciences. In the revised version, Authors have properly addressed all of my comments and questions. Furthermore, after revision and providing of the proper explanations, the quality of the manuscript has been significantly improved. In my opinion, presented manuscript might be accepted for publication in the International Journal of Molecular Sciences in the present form.
Reviewer 2 Report
The authors have substantially improved the manuscript. They have corrected the errors and have taken into consideration the suggestions of the reviewers. Therefore, I would like to recommend this manuscript for publication in IJMS.
This manuscript is a resubmission of an earlier submission. The following is a list of the peer review reports and author responses from that submission.
Round 1
Reviewer 1 Report
Presented manuscript concerns selection of seven Lactobacillus plantarum strains isolated from Malaysian foods and production of the extracellular proteolytic, cellulolytic and hemicellulolytic enzymes form these strains. Further, obtained strains were used and compared for biotransformation of palm kernel cake biomass. In the study, activity of the produced enzymes was evaluated as well as effect of pH on this parameter was examined.
The topic discussed in the manuscript is definitely worth of investigation and is within the scope of the International Journal of Molecular Sciences. The manuscript is nice planned and organized and might be interesting for a broad variety of readers. Nevertheless, in my opinion, before manuscript might be considered for publication in IJMS some improvements and explanations are required. For this reason, see the detailed comments presented below.
In the whole manuscript discussion and interpretation of the obtained results and obtained phenomena is insufficient and should be significantly improved. It should be clearly presented how obtained results affect properties of the microbial strains and further study. Moreover, there is a lack of proper results interpretation form the point of view of chemistry and microbiology. In this aspect, the whole manuscript should also be significantly improved.
Novelty of the study should be clearly presented in the Introduction section.
Section 2.1.2, Figure 2. Brief description of the way how the results presented in Fig. 2 were obtained should be addedd in this section. Moreover, there is a lack of data interpretation and information how the obtained data might be use.
Please improve Figure 3 caption, as some names of the bacterial strains are unnecessary.
Not only activity but also amount of the produced enzymes after the fermentations should be provided in the manuscript.
Images presented in Figs. 5 and 6 should be enlarged.
How changes in the enzymatic activity of all of the tested enzymatic groups might be explained? Moreover effect of the pH on enzyme activity should be discussed and commented more detail.
Section Discussion could be divided into subsections.
I encourage Authors to present some of the results demonstrated surface properties of the LAB, as these parameters significantly affect production of the enzymes.
Pages numbering in the whole manuscript should be unified.
Materials and Methods, Section 4.7. this section should be presented more detail.
Conclusions section should be presented more detail. For instance some of the results should be summarized as well as future trends in the presented research area should be highlighted.
References list required some minor improvements.
I encourage Authors to consult native speaker to check the English in the manuscript.
Reviewer 2 Report
This manuscript shows the study of the hydrolytic activities from seven Lactobacillus plantarum strains. The authors explored the cellulolytic, proteolytic and hemocellulolytic enzyme activities. Thus, the selected L. plantarum strains were investigated for the hydrolysis of palm kernel cake biomass via solid-state fermentation. The results shown here could be of interest of a wide audience working in biocatalysis.
My comments are as follows:
What is the composition of the palm kernel cake extract?
What reducing sugars are produced? I.e. what is the composition of the products after hydrolysis?
Could the protease activity be hydrolyzing cellulases and hemicellulases? If this were the case, it would affect the production of simple fermentable sugars.
Line 150. Please correct this paragraph. As shown in figure 1f, isolate B4 produce more than 3 U/mg of protease activity at pH 6.5. While at pH 5, B4 produce less than 1 U/mg.
Have the authors considered analysing the hydrolytic cocktail from L. plantarum by means of for example LC-MS/MS?
Compared to the hydrolytic cocktails produced by fungus, how active are the cocktails produced by L. plantarum?